# Surgical Management of Pre-Chiasmatic Intraorbital Optic Nerve Gliomas in Children after Loss of Visual Function—Resection from Bulbus to Chiasm

**DOI:** 10.3390/children9040459

**Published:** 2022-03-24

**Authors:** Julian Zipfel, Jonas Tellermann, Dorothea Besch, Eckart Bertelmann, Martin Ebinger, Pablo Hernáiz Driever, Jens Schittenhelm, Rudi Beschorner, Arend Koch, Ulrich-Wilhelm Thomale, Martin Ulrich Schuhmann

**Affiliations:** 1Section of Pediatric Neurosurgery, Department of Neurosurgery, University Hospital of Tuebingen, 72076 Tuebingen, Germany; jonas.tellermann@med.uni-tuebingen.de (J.T.); martin.schuhmann@med.uni-tuebingen.de (M.U.S.); 2Centre of Neurofibromatosis, Centre of Rare Diseases, University Hospital of Tuebingen, 72076 Tuebingen, Germany; 3Section of Periocular and Orbital Surgery, Department of Ophthalmology, University Hospital Tuebingen, 72076 Tuebingen, Germany; dorothea.besch@med.uni-tuebingen.de; 4Department of Ophthalmology, Charité-Medical University, 13353 Berlin, Germany; eckart.bertelmann@charite.de; 5Department of Pediatric Oncology, University Children’s Hospital of Tuebingen, 72076 Tuebingen, Germany; martin.ebinger@med.uni-tuebingen.de; 6Department of Pediatric Oncology/Hematology and Stem Cell Transplantation, Charité-Medical University, 13353 Berlin, Germany; pablo.hernaiz@charite.de; 7Institute of Pathology and Neuropathology, University Hospital of Tuebingen, 72076 Tuebingen, Germany; jens.schittenhelm@med.uni-tuebingen.de (J.S.); rudi.beschorner@med.uni-tuebingen.de (R.B.); 8Institute of Neuropathology, Charité-Medical University, 13353 Berlin, Germany; Arend.koch@charite.de; 9Department of Neurosurgery, Division of Pediatric Neurosurgery, Charité-Medical University, 13353 Berlin, Germany; ulrich-wilhelm.thomale@charite.de

**Keywords:** optic pathway glioma, neurofibromatosis type 1, exopthalmus, surgical management, intraorbital surgery, intra-extradural surgery

## Abstract

Optic pathway gliomas in children carry significant morbidity and therapeutic challenges. For the subgroup of pre-chiasmatic gliomas, intraorbital and intradural resection is a curative option after blindness. We present a two-center cohort using different surgical approaches. A retrospective analysis was performed, including 10 children. Mean age at surgery was 6.8 years. Interval between diagnosis and surgery was 1–74 (mean 24 ± 5.5, median 10) months. Indications for surgery were exophthalmos, pain, tumor progression, or a combination. Eight patients underwent an extradural trans-orbital-roof approach to resect the intra-orbital tumor, including the optic canal part plus intradural pre-chiasmatic resection. Gross total resection was achieved in 7/8, and none had a recurrence. One residual behind the bulbus showed progression, treated by chemotherapy. In two patients, a combined supra-orbital mini-craniotomy plus orbital frame osteotomy was used for intraorbital tumor resection + intradural pre-chiasmatic dissection. In these two patients, remnants of the optic nerve within the optic canal remained stable. No patient had a chiasmatic functional affection nor permanent oculomotor deficits. In selected patients, a surgical resection from bulb to chiasm ± removal of optic canal tumor was safe without long-term sequela and with an excellent cosmetic result. Surgery normalizes exophthalmos and provides an effective tumor control.

## 1. Introduction

Optic pathway gliomas (OPG) are rare tumors in pediatric patients with challenges concerning diagnostics and therapy. The group of OPG include a vast range of clinical manifestations as well as anatomical sites of manifestation [1,2]. Tumors may arise from and involve the optic nerve only, the optic chiasm and hypothalamus as well as the postchiasmatic optic tract, or affect the whole optic pathway diffusely. Whilst symptomatology is broad, the most common morbidity of these tumors is visual impairment in about 30% of cases. Histology is mostly pilocytic astrocytoma [3,4]. 

In the vast majority of manifestations, watchful waiting, or—in cases of visual impairment/deterioration or relevant tumor growth—chemotherapy is the treatment modality of choice. As long as there is useful vision, surgical intervention should be limited to the resection of relevant tumor cysts, partial resection of exophytic parts, central tumor debulking to decrease the pressure on the surrounding visual or hypothalamic structures, or diversion of cerebrospinal fluid in cases of hydrocephalus [5].

There is an association with neurofibromatosis type 1 (NF1), where about 15% of patients develop OPGs. NF1-associated tumors tend to be less clinically aggressive and relevant, and consequently only 2–5% of NF1 patients are treated for OPG [6,7], whilst female patients may be affected more severely [8].

A special entity of OPG are gliomas of the prechiasmatic optic nerve without involvement of the chiasm itself. These tumor manifestations can be intradural only, intra-orbital only, or in both compartments. The natural history of these tumors may show progression in up to 60%, with spontaneous regression in 18%, the latter only in association with NF1. A visual decline has been reported in about half of patients [5]. In pre-chiasmatic OPG, a curative treatment via surgery is feasible in cases of blindness or loss of functional vision [9]. Especially in patients with a significant intraorbital tumor mass causing exophthalmos and sometimes orbital pain, surgery is an excellent option, since, in contrary to chemotherapy or radiation, exophthalmos and pain are immediately removed and a long chemotherapy or long-term sequalae of radiation in children is avoided. A prechiasmatic transection of the affected optic nerve can prevent tumor in-growth into the chiasm. Surgery, however, must preserve oculomotor function as well as cosmetic integrity. Amongst others, pterional approaches have been proposed [4]. The largest cohort of OPG of the optic nerve has been reported by Shofty et al., who proposed surgical intervention in selected cases [5].

In this study, we present a two-center cohort of pediatric patients with prechiasmatic tumor manifestation, exophthalmos, and loss of functional vision using two different surgical approaches. We describe indication and surgical techniques, and analyze the long-term outcome.

## 2. Materials and Methods

A retrospective analysis of the pediatric patient databases of the Divisions of Pediatric Neurosurgery at the University Hospital of Tuebingen and the Charité Berlin was performed to identify patients younger than 18 years at the time of diagnosis with isolated prechiasmatic optic nerve glioma and significant intraorbital tumor mass, in whom a surgical resection from the bulbus to the chiasm was performed. In Tuebingen and Berlin, a different surgical approach and strategy was applied. 

Indications for surgery were blindness/non-serviceable vision of the affected eye, disfiguring exophthalmos, pain, and documented tumor growth. A combination of factors was often present. All preoperative MRIs showed contrast-enhancing tumor growth from the orbit into the optic canal, and in 8/10, further intradural tumor extension affecting the ipsilateral optic nerve but not the optic chiasm (Figure 1). 

Chart review and retrospective analysis of discharge and follow-up reports, ophthalmological testing, and neuropathology reports (including methylation array) was performed to acquire information on patient history, symptomatology, outcome, and further treatment.

### 2.1. Quantification of Exophthalmos

Clinically visible exophthalmos was assessed via MRI scans and quantification, as described previously [10].

### 2.2. Statistical Analysis

Statistics were analyzed using SPSS Statistics 25 (IBM, Armonk, NY, USA). Continuous data were presented as mean (± standard deviation), whereas categorical data were shown as percentages. Continuous variables were tested for equality of variances by Levene’s test. Normal distributed parametric variables with equal variances were compared using the unpaired or paired *t*-test, otherwise the Mann–Whitney *U* test was performed. Nominal variables were tested with Fisher’s exact test. *p* values < 0.05 were regarded as significant.

This study was performed in line with the principles of the Declaration of Helsinki. Institutional board approval was granted by the Ethics Committee of University of Tuebingen (762/2021BO2; 2 December 2021). Written informed consent has been obtained from the patients’ parents to publish the photographs in this manuscript.

### 2.3. Surgical Approaches

In Tuebingen (Group A), a fronto-temporal skin incision was used, followed by supraorbital craniotomy. After epidural exposition of the orbit, its roof is opened by osteotomy and the optic canal is unroofed with a diamond drill back to the dural fold. The anterior clinoid process is not drilled, and care is taken not to open the ethmoidal cells medial of the optic canal. After T-shape opening of the periorbital fascia, the frontal branch of the first trigeminal nerve and the superior rectus muscle/levator palpebrae muscle are dissected and retracted laterally, including the superior oculomotor branches entering the muscle from below, in order to expose the tumor. The abducens nerve is not encountered during this approach.

The nasociliary nerve crossing the optic nerve in the middle of the orbit and the trochlear nerve crossing close to the conus are identified to avoid injury during resection. Then, the dura of the optic nerve sheath covering the tumor is opened and an intradural tumor resection is performed, directed anteriorly following the nerve sheath, until the eye bulb is reached. Inside of the dura, no relevant neurovascular structures are encountered. Great care is taken not to injure the eye bulb at the level of optic nerve transection as close to the bulbus as possible. Then, the tumor is followed posteriorly within the optic nerve sheath towards the optic canal. The muscle attachment of the superior muscles at the orbital cone needs to be spared as well as injury to the visible trochlear nerve avoided. Then, the intradural resection is continued behind the muscle attachment within the optic canal towards the intracranial cavity until CSF comes from the intradural compartment. At this point, the dura is opened in a straight line at the level of the skull base; the intradural optic nerve, chiasm, and carotid artery are dissected; and the optic nerve is transected between the visible end of the tumor and chiasm. In the case of tumor extension close to or immediately at the chiasm, care must be taken not to enter the latter to avoid any injury of crossing fibers. The remaining tumor part in the junction area of the extradural optic canal and the intradural compartment just above the carotid and ophthalmic artery is carefully mobilized from its attachments using the intra- and extradural route and finally removed. The ophthalmic artery can be visualized in the first millimeter intradurally; the dissection at this transition zone has to be performed very carefully and without force, and this way injury to the artery is avoided. The artery can also be seen at the very beginning of the optic canal after tumor removal, but mainly runs behind the dural optic nerve sheath, and is thus not endangered there, just as the inferior oculomotor branches intraorbitally. The optic canal is plugged with muscle fixed with fibrin glue, and then the periorbital fascia is reconstructed with 6-0 sutures. The bone of the orbital roof is replaced and fixed with PDS sutures, and the dura closed thereafter. The craniotomy is refixed with PDS sutures in children below 15 years, otherwise titanium plates are used. Intraoperative photographs are provided in Figure 2. 

In Berlin (Group B), a lateralized eyebrow incision was applied to expose the supraorbital rim after incision of the orbicularis muscle and periosteal and anterior parts of the temporal muscle dissection. Using a piezosurgery craniotome, a supraorbital mini-craniotomy was performed by leaving the supraorbital nerve medially intact and including the zygomatic process of the frontal bone. The craniotomy exposed the periorbital as well as the frontal basal dura. After scleral incisions, the rectus muscles were identified and ligated to achieve control over the muscular structures during tumor resection. The periorbit was opened, and intraorbital structures were secured as described above. The intraorbital fat was reduced and retracted to prepare for dissection towards the optic nerve under ultrasound guidance. The optic nerve sheath was opened, and the tumor was debulked by ultrasound aspiration. After sufficient space was gained, the posterior bulbus and the insertion of the optic nerve was identified. The nerve was cut in close proximity to the bulb. Then, the remnants of the nerve and the tumor were dissected towards the orbital cone. Just in front of the optical canal, the most posterior segment of the optic nerve was dissected. After hemostasis, the periorbit was sutured using 5-0 vicryl. Then, the dura was opened, and a subfrontal approach was used to identify the intracranial pre-chiasmatic part of the optic nerve. This part was coagulated and cut, leaving a few millimeters of the nerve not to harm the chiasm. The dura was closed in a watertight fashion. The supraorbital bone-flap was reinserted and fixated with titanium microplates. After orbicularis adaptation and subcutaneous sutures, the skin was glued. The rectus muscle ligations were withdrawn, and the sclera was sutured. 

## 3. Results

A total of 77 children with OPG were identified in our Tuebingen database, of whom eight consecutive children fulfilled the inclusion criteria with unilateral prechiasmatic OPG. Two additional patients from Berlin with supraorbital mini-craniotomy were included. Patients were predominantly female (*n* = 6, 60%). Mean age at diagnosis was 4.4 years (range 0–16 years), and mean and median ages at surgery were 6.4 ± 6.5 and 3.5 years, respectively. Table 1 shows the basic patient characteristics.

Six tumors were left-sided (60%), and the other four were on the right side. Mean time to diagnosis from first symptoms was mean 24 ± 5.5 (median 10) months. NF1 was present in four children (40%). 

Histology revealed pilocytic astrocytoma in all cases. A molecular genetic panel analysis and methylation array was available for 8/10 cases, and the results are summarized in Table 2 (see Figure 3 for exemplary histological cross-section).

There was no uniform pattern of underlying molecular characteristics associated with this type of tumor manifestation. All tumors except one case clustered to methylation class low-grade glioma, including subclassification midline pilocytic astrocytoma in four cases. One case clustered into the basket group low tumor content/reactive tissue.

### 3.1. Previous Therapy

A total of 5 out of 10 children had previous therapies. One child had received chemotherapy alone, another child chemo- and radiotherapy. Both showed recent tumor progression and increasing exophthalmos. 

One patient (#1) had chemotherapy and bony orbital decompression of orbital canal and lateral orbital wall. Both measures did not prevent blindness, and further tumor progression led to further exophthalmos and strong pain in the orbit.

One child (#3 Table 1) had previous intraorbital surgery at another institution. The child was diagnosed with optic nerve glioma because of exophthalmos aged 10 months and treated for 18 months with chemotherapy. At tumor progression one year later, she underwent a partial resection of the intraorbital tumor via a trans-orbital roof approach and transection of the intradural pre-chiasmatic tumor. The remaining tumor showed early progression, and 5 months later, the eye was enucleated, including the anterior parts of the tumor-affected optic nerve. The remaining tumor in the optic canal and the cone region showed progression 7 months later, and the child was scheduled for radiation when she presented to our institution for a second opinion. The parents decided for total tumor resection instead of radiation in a 4.5-year-old child.

One patient (#7, Table 1) aged 17 with active and problematic ulcerous colitis presented with predominant unilateral visual impairment due to one-sided intradural pre-chiasmatic plus chiasmatic tumor. She initially received a partial prechiasmatic tumor debulking and proton therapy of the intradural manifestation. Vision stabilized bilaterally for about 5 years, until she lost vision on the predominantly affected side and developed exophthalmos due to intraorbital tumor progression. The chiasmatic tumor and vision of the other eye remained stable. Since another radiation treatment was neither desired by the patient nor possible, the indication for bulbus to chiasm resection was given at age 21 years.

### 3.2. Symptomatology

Initial symptoms were predominantly loss of vision (*n* = 10, 100%), followed by exophthalmos in seven children (70%) and pain in one child. Another child developed secondary pain. The interval between diagnosis and surgery was 1–74 (mean 24 ± 5.5, median 10) months. Indication for surgery was significant, with progressing exophthalmos alone in one child, tumor growth with or without progression of exophthalmos in eight children (80%), and pain in another child. 

### 3.3. Exophthalmos

In 8/10 cases (80%), clinically visible exophthalmos was present at the time of surgery. On MRI, exophthalmos, defined as ∆ > 1 mm to the other side, ranged from 3.2 to 13.0 mm (mean 5.3 ± 3.7 mm). Postoperatively, no exophthalmos was present in any case, and in 2/10 cases, a slight enophthalmos with a ∆ of −2.36 and −2.26 mm existed. Both cases were status after pre-operative radiation of the orbital tumor. 

### 3.4. Ophthalmologic Status

At the time of surgery, there was in all cases either total blindness without light reflex or non-functional visual perception. Postoperatively, there was no impairment of the visual field or acuity of the remaining eye, and thus no affection of the chiasmatic function in any patient old enough to undergo visual field examination. The two patients that received radiotherapy prior to surgery showed affection of the levator palpebrae function postoperatively with a mild-to-moderate degree of ptosis. 

### 3.5. Surgery

Gross total resection (GTR) was achieved in 7/8 cases in Group A (87.5%). Not intended subtotal resection (STR) occurred in the first case of our series, wherein a small residual remained in the dural sleeve immediately adjacent to the bulbus. In the Berlin series (Group B), the optic nerve without clear tumor involvement in preoperative MRI remained within the optic canal intendedly.

No intraoperative or immediate post-operative complications were observed, and patients were usually discharged with orbital swelling and not yet or returning levator palpebra function. In the follow-up, two children developed enophthalmos, one with minimal levels and one with moderate ptosis due to levator palpebra muscle weakness. Both children had received radiotherapy prior to surgery. The other eight (80%) children had no obvious oculomotor or cosmetic deficit of eye position and movement. Figure 4 shows an example of resolution of exophthalmos on MRI after tumor removal.

Figure 5 provides exemplary photographs of a child with good oculomotor function two years postoperatively.

Figure 6 provides exemplary photographs of a 2-year-old child with a severe preoperative exophthalmos and good cosmetic result 13 days postoperatively.

### 3.6. Follow-Up

The mean follow up was 42.0 (range 2–74) months in Group A, and 15 months in Group B. Patients received MRIs at 3 months post-surgery, then in cases of no residual tumor, every 6 to 12 months, depending on previous tumor growth velocity for up to 5 years. The case with unintended STR (No 1, Table 1) showed tumor progression in imaging and was subsequently treated with chemotherapy. No further growth was observed thereafter. In the two cases from Group B, there was stability of the intended residual optic nerve within the optic canal and no signs of tumor recurrence originating from there. In two of the four cases with NF1, a very small contralateral optic nerve glioma (MRI signal intensity changes and minimal contrast uptake, but no significant tumor extension) was known prior to surgery. Since in both children, no impairment of vision (visus 0.8 and 1.0) existed, this tumor manifestation was only observed in accordance with guidelines that asymptomatic tumors do not deserve treatment. An exemplary MR image set is provided in Figure 7.

## 4. Discussion

In this study, we report on a cohort of pediatric patients with unilateral mainly prechiasmatic optic nerve gliomas with progressive intraorbital tumors, progressive exophthalmos in most, and pain. All patients had lost visual function. In such selected cases, surgery removing the tumor entirely from the bulbus to the chiasm is possible and feasible, solving the problem of pain and exophthalmos at once without significant morbidity and a good cosmetic outcome. In cases of a tumor only within the orbit, the unaffected optic nerve within the optic canal can remain. Major theoretical risks involve lesion to oculomotor nerve branches, leading to ptosis or eye axis deviation, lesion to the ophthalmic artery, and injury to the bulb. Indication for surgery is limited to cases with blindness or non-serviceable vision and progressing exophthalmos or pain, as well as radiologically progressing tumor. This is a small group considering all cases with optic pathway gliomas, but one where surgical therapy provides a cure. 

The largest previously published case series on surgical interventions in children for this type of surgery (*n* = 4) utilized a pterional (more lateral) approach and resection of the lateral orbital wall plus clinoid process plus parts of the roof to display the optic nerve extra- and intradurally [4]. We describe two different approaches.

The advantages of the trans-orbital roof approach to the orbit from above compared to a pterional approach are that the temporal muscle does not need to be mobilized and incised, thus with less postoperative pain and discomfort when chewing and zero risk for affection of the frontal branch of the facial nerve during the approach. Furthermore, no cosmetic problems can arise from temporal muscle atrophy, which may present in up to 16% of patients after pterional craniotomy [11].

Moreover, there is a better visualization of the transition of the tumor from the orbit into the optic canal, without the need of exposing the superior orbital fissure and performing an anterior clinoidectomy. This decreases the surgical risks considerably. Further advantages are the option for excellent reconstruction of the periorbita and of the orbital roof, which can be totally replaced after initial osteotomy.

The advantages of the transorbital approach after orbital rim removal and mini-supraorbital craniotomy are a smaller skin incision with an extended eyebrow incision. The obtained exposure is sufficient to remove all intraorbital tumor parts. However, the viewing angle and the exposure does not allow for the removal of existing tumor parts from the conus onwards into the optic canal. The intracranial prechiasmatic optic nerve was removed in order to prohibit possible tumor invasion into the chiasm in case there was microscopic tumor involvement of this part of the optic nerve that was invisible on MRI. Still, a safety margin has to be considered as unilateral optic nerve gliomas limited to the prechiasmatic nerve on MRI may extend beyond the MRI borders [12]. Therefore, this approach seems to be well suited for cases with only intra-orbital tumor manifestation, as was the case in both cases described herein, where imaging did not show clear tumor involvement of the intracanalicular and pre-chiasmatic part of the optic nerve. Since in the two cases, no progression or recurrence of the tumor was seen during a mean follow-up of 15 months, it seems that this approach is a veritable alternative, especially if the tumor is strictly located in the intraorbital optic nerve portion.

Usually, OPG are not primary candidates for surgical resection, and mostly chemotherapy is the treatment of choice. Indication for treatment arises from progressive visual impairment and not from radiological progress or change in contrast uptake [13,14]. Carboplatin/Vincristin has been the standard therapy in the past. Currently, Vinblastin monotherapy is an alternative option with less side effects [15]. Especially in NF1 patients, there is a rationale for the more recently available MEK inhibitors, but consistently large studies in OPGs are still missing [16].

Two of the children in this series with NF1 had a visible tumor in the contralateral optic nerve with increased signal intensity of the nerve and mild contrast uptake. The visual acuity of this nerve was unimpaired. In both cases, coming from outside institutions, an argument was made in the primary treating units that the existence of the contralateral tumor demanded chemotherapy instead of surgery, because of the risk of visual loss on the other side if chemotherapy would not be administered initially. In our opinion, however, the main problem in these children was a progressive tumor with exophthalmos. This issue of tumor mass will not be addressed by chemotherapy. Furthermore, the pure existence of an optic glioma in NF1 children is not an indication for chemotherapy per se, as long as there is no visual impairment. This remains true also in the case that there is a contralateral progressive tumor that leads to blindness, because bilateral tumors can and tend to behave quite differently on either side, and proof of visual impairment needs to exist prior to long-lasting chemotherapy [17]. In both cases in this series, the contralateral tumors remained silent during follow-up (4 and 10 years, respectively), and no visual impairment or radiological progression was observed.

The molecular analysis showed no uniform molecular pattern associated with progressive intraorbital tumors. However, the first two samples have been processed with the older 450 K methylation array and older classifier than the later ones, and thus the result was unspecific and did not include additional sequencing data. With the new panel sequencing techniques, there might have been more detailed results. On the other hand, molecular profiling in NF1 patients with and without associated optic gliomas did not reveal certain underlying genetic patterns with the optic glioma group [18]. The last two cases showed FGFR1 hotspot mutations previously reported in six optic gliomas [19]. We have very recently demonstrated that FGFR1 hotspot mutations are also frequently seen in other low-grade tumor entities [20]. Therefore, it is unlikely that a certain molecular subtype constitutes a risk factor for development of pre-chiasmatic and especially intraorbital progressive tumors.

Radiotherapy is usually not recommended in children if alternatives exist, and especially in NF1 children, it should be avoided due to increased long-time risk [21]. Proton beam radiation has been performed, but radiological follow-up may be difficult to interpret due to pseudoprogression [22]. As our series shows, there are side effects of radiotherapy on the orbit, since only those patients developed enophthalmos and ptosis.

## 5. Conclusions

The transfrontal combined intra- and extradural approach from above and the transorbital/minimal supraorbital approach from the front are safe and feasible surgical techniques for complete resection of at least the intraorbital optic nerve gliomas and the intradural pre-chiasmatic tumor. The first approach in addition also enables the total resection of the parts within the optic canal and the transition zone to the intradural compartment. In experienced hands with an interdisciplinary team from pediatric neurosurgery and ophthalmology, morbidity is low and curative treatment is possible in selected patients, preventing at least one year of chemotherapy in smaller children or radiation therapy in older ones. In contrary to non-surgical therapy, surgery removes exophthalmos and pain immediately with excellent cosmetic outcome regarding eye movement and eye lid elevation, especially if no radiotherapy was performed earlier. For the described patient subgroup of pediatric optic pathway gliomas, the described surgical therapy seems to be the method of choice.

## Figures and Tables

**Figure 1 children-09-00459-f001:**
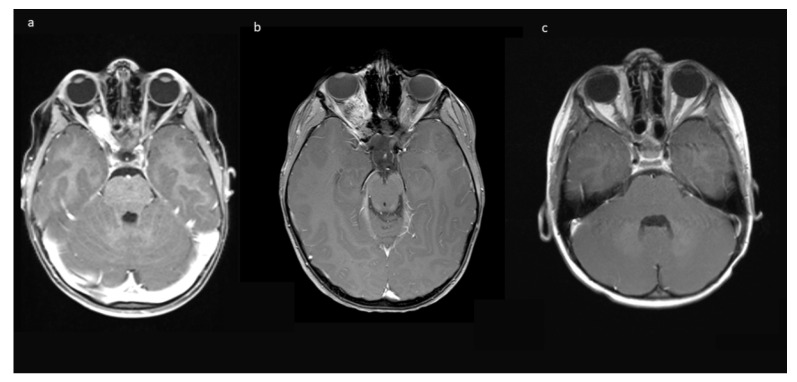
Exemplary MR-imaging of a 3-year-old male (**a**) preoperatively, (**b**) 1 month postoperatively, and (**c**) 2 years postoperatively.

**Figure 2 children-09-00459-f002:**
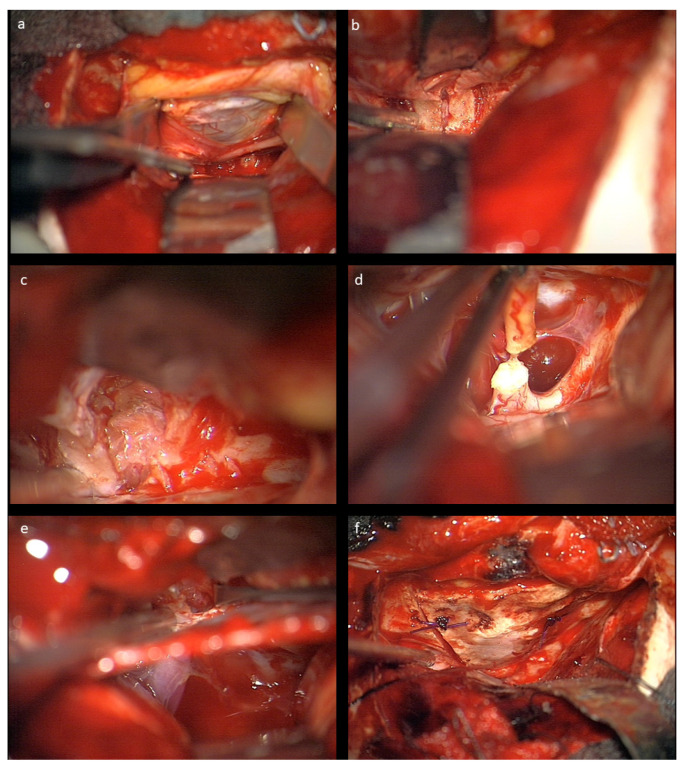
Intraoperative microscopic view after left-sided craniotomy: (**a**) The optic roof was removed and the periorbita is visualized; the frontal nerve (V1) is seen under the retractor. (**b**) The optic canal is visualized extradurally. (**c**) Resection of the glioma in the optic nerve sheath intraorbitally. (**d**) Transection of the optic nerve prechiasmatically. (**e**) Combined intra- and extradural visualization of the optic canal and the intradural space medial to the carotid aretery. At the bottom of the optic canal, the beginning of the ophthalmic artery can be seen. (**f**) Reconstructed orbital roof.

**Figure 3 children-09-00459-f003:**
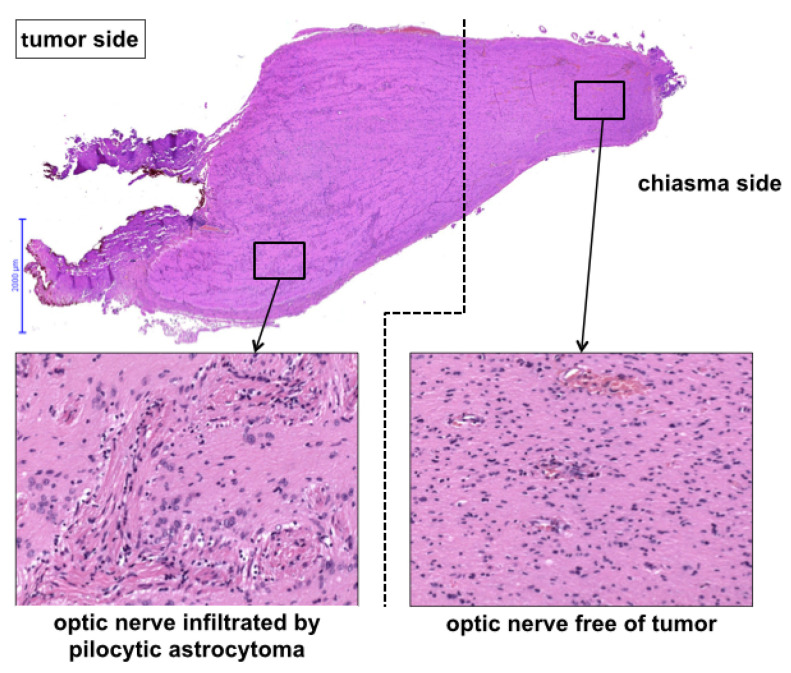
Histological view of the pre-chiasmatic tumor margin.

**Figure 4 children-09-00459-f004:**
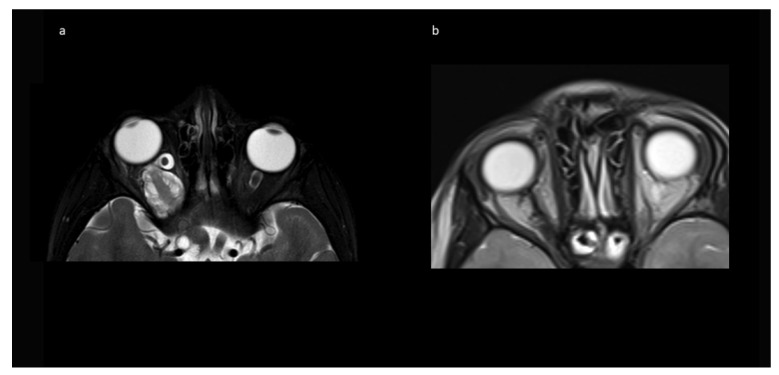
Exemplary MR imaging follow-up of a child with (**a**) preoperative exophthalmos on the left side and (**b**) postoperative normal eye position.

**Figure 5 children-09-00459-f005:**
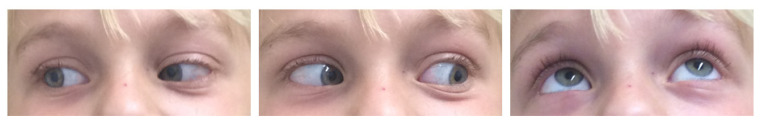
Exemplary photographs 2 years postoperatively with intact oculomotor function.

**Figure 6 children-09-00459-f006:**
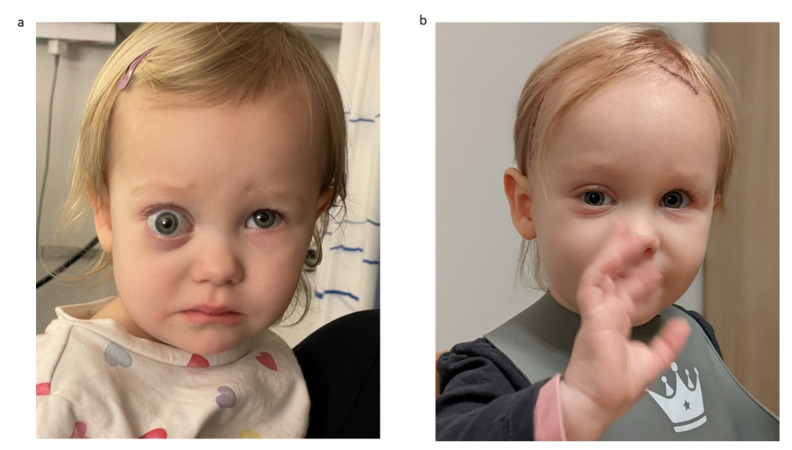
Exemplary photographs of a 2-year-old child with (**a**) preoperative status with severe exophthalmos and (**b**) status 13 days postoperatively with a good cosmetic result and normal eye position.

**Figure 7 children-09-00459-f007:**
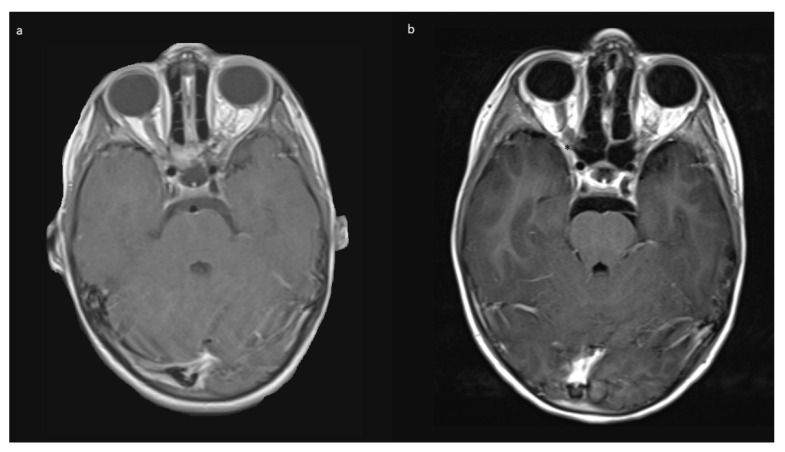
Exemplary MR imaging follow-up of a small contralateral right-sided prechiasmatic optic nerve glioma (**a**) 12 months postoperatively; (**b**) 60 months postoperatively.

**Table 1 children-09-00459-t001:** Basic patient characteristics.

Patient no.	Age at Diagnosis (Years)	Sex	Side	Preoperative Therapy	NF1	Preoperative Vision	Indication for Surgery	Quantification of Exophthalmos (mm)	Surgical Approach (Group)	GTR/STR	Follow-up (Years)
#1	3	female	left	bony surgery, chemotherapy	yes	0.0, optic atrophy	pain, exophthalmos	7.32	A	STR	1
#2	3	female	left	-	yes	0.0, optic atrophy	pain	3.73	A	GTR	6
#3	3	female	left	chemotherapy, surgery	no	0.0	tumor progression	3.97	A	GTR	4
#4	5	female	left	chemotherapy, radiation	no	0.0	tumor progression	6.44	A	GTR	4
#5	3	male	right	-	no	0.0	tumor progression	3.24	A	GTR	10
#6	3	male	right	-	yes	0.0	tumor progression	0	A	GTR	4
#7	16	female	right	radiation	no	0.1, optic atrophy	tumor progression	0	A	GTR	3
#8	2	female	right	-	yes	blind	exophthalmos, tumor progression	4.04	A	GTR	0.1
#9	5	male	left	chemotherapy	no	0.0	tumor progression, exophthalmos	na	B	STR	0.8
#10	6	female	left	-	no	0.0	tumor progression, exophthalmos	na	B	STR	1.7

NF, neurofibromatosis; GTR, gross total resection; STR, subtotal resection.

**Table 2 children-09-00459-t002:** Results of molecular genetic analysis.

Patient No.	Methylation Classifier Score/Diagnosis (850k)
#1	0.426 low-grade glioma (450k array, classifier V5.1)
#2	0.414 low-grade glioma (subtype midline, 450k array classifier V8.0)
#3	0.31 low-grade glioma (subtype midline PCA), KIAA1549-BRAF Fusion
#4	0.45 low tumor content, reactive tissue
#5	0.30 low-grade glioma (subtype midline PCA), no KIAA1549-BRAF fusion, NF1 frameshift deletion, EGFR nonsynonymous SNV
#6	0.37 low-grade glioma (subtype midline PCA)
#7	1.0 low-grade glioma (subtype midline PCA), FGFR1 hotspot mutation
#8	1.0 pilocytic astrocytoma (subtype midline PCA), FGFR1 hotspot mutation

PCA, pilocytic astrocytoma; NF, neurofibromatosis; EGFR, epidermal growth factor receptor; SNV, single nucleotide variant; FGFR, fibroblast growth factor receptor.

## Data Availability

Data are available from the corresponding author.

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
