# Peer review of "Surgical Management of Pre-Chiasmatic Intraorbital Optic Nerve Gliomas in Children after Loss of Visual Function—Resection from Bulbus to Chiasm"

_children, 2022, doi:10.3390/children9040459_

Round 1
Reviewer 1 Report
The authors describe a cohort of pediatric patients with unilateral prechiasmatic optic nerve gliomas with progressive intraorbital tumors, progressive exophthalmos and pain.
The manuscript deal with the matter of surgery for optic pathway gliomas in a well-balanced manner, focusing on the feasibility of surgery in such selected cases of prechiasmatic gliomas. The transection of the optic nerve occurs by means of two surgical approaches, both without resection of the lateral orbital wall and without anterior clinoidectomy. These approaches are a real alternative to the pterional approach for tumors located in the intraorbital optic nerve portion and only with intraorbital manifestations. Furthermore, it would be useful to further specify how to avoid lesion to oculomotor nerve branches, ophthalmic artery and bulb.
Nothing innovative about molecular genetics.
Table and figure legends are explanatory and fairly clear, and the paper includes adequate references. English language and style are fine.
Reviewer 2 Report
This is a very interesting article about the surgical management of optic pathway gliomas in children using different surgical approaches.
- This is an important article as it explores an entity that can be very symptomatic but is also treatable.
- Line 79: the word "indentifie" may be misspelled
- The surgical method is well explained and the differences between the surgical methods was well contrasted for specific situations.
- the comparison with other methodologies of treatment was a good comparison.
- line 317 needs revision
- This is a well written paper that gives alternatives to treatment with chemotherapy/radiation
